# P Starvation in Roses Leads to Strongly Genotype-Dependent Induction of P-Transporter Genes during Black Spot Leaf Disease

**DOI:** 10.3390/jof8060549

**Published:** 2022-05-24

**Authors:** Helena Sophia Domes, Enzo Neu, Marcus Linde, Thomas Debener

**Affiliations:** 1Department of Molecular Plant Breeding, Institute for Plant Genetics, Leibniz Universität Hannover, 30419 Hannover, Germany; domes@genetik.uni-hannover.de (H.S.D.); linde@genetik.uni-hannover.de (M.L.); 2Julius Kühn-Institut, Federal Research Centre for Cultivated Plants, Institute for National and International Plant Health, 38104 Braunschweig, Germany; 3KWS SAAT SE & Co. KGaA, 37574 Einbeck, Germany; enzo.neu@kws.com

**Keywords:** abiotic stress, crosstalk, gene expression, genotypic effect, phosphate starvation, phosphate transporter, plant defence, plant pathogen, RNA-seq

## Abstract

Phosphorous starvation in plants has been reported to have contrasting effects on the interaction with pathogens in different plant pathogen systems and plant species. Both increases and decreases in susceptibility have been observed in numerous reports. Here, we analysed black spot infection and the leaf expression of two plant phosphate transporters and one defence marker gene in roses after phosphorous starvation. We varied three factors: phosphate starvation versus full supply of phosphorous, black spot infection vs. mock inoculation, and different susceptible and resistant progeny of a biparental rose population. Black spot susceptibility or resistance was not significantly changed upon phosphate starvation in either compatible or incompatible interactions. The expression of phosphate transporters was strongly induced upon starvation, but in some genotypes, expression was altered by black spot interaction as well. The marker for pathogenic interactions was exclusively induced by interaction with black spot, but the expression was altered by a combination of phosphate starvation and interaction with the fungus in some genotypes. In summary, phosphate starvation has clear effects on the gene expression of phosphate transporters in rose leaves, and the interaction with a hemibiotrophic leaf pathogen is strongly genotype dependent.

## 1. Introduction

Phosphorus (P) is a macronutrient that plays an important role in many plant physiological processes. However, the natural occurrence of inorganic phosphate (Pi), the most common uptake form of P, is usually so limited that deficiency is widespread, especially in plants due to their sessile lifestyle (e.g., [1]). Thus, concentrations of more than 10 µM Pi [2] rarely occur in soils (usually 2 µM), whereas the concentration in plant tissue is 5–20 mM [1].

To compensate for this, plants have developed various strategies. On the morphological level, there are mainly adaptations of the root system to Pi deficiency. These include an increased root:shoot ratio [3], the proliferation of root hairs [4], and an increased association with mycorrhizal fungi [5]. At the physiological level, the plant redistributes available Pi inside the plant and secretes organic acids to increase the availability of Pi within the rhizosphere [6,7]. At the biochemical level, a phosphate deficiency can lead to an adjustment of enzyme levels (e.g., [8]) and an increased production of phosphatases [9]. Another adaptation comprises an increase in the number of phosphate transporters (PHTs). Transporters are necessary to overcome the Pi concentration gradient between the soil and cell and, with increasing abundance, can further enhance Pi uptake; they are also needed for distribution within the plant.

Various families of PHTs perform different tasks in plants. In the uptake of Pi from the soil, members of the phosphate transporter family 1 (PHT1) play a major role. They are located in the plasma membrane, and their number is increased under phosphate deficiency (e.g., [10,11]). However, aside from some well-studied plants, such as *Arabidopsis* and rice, knowledge about the members of the phosphate transporter families and their regulation is limited. In the Rosaceae family, PHTs have thus far only been characterised in detail for *Malus domestica* [12].

More complex reactions were observed when several stress factors occurred at the same time. Investigating how signalling pathways can influence each other is an important step towards understanding how processes take place under natural conditions [13]. In an in silico analysis of *AtPHT* promoter regions, many possible regulatory elements were found that indicate crosstalk with other signalling pathways [14], including phytohormones and both abiotic and biotic stress signalling.

A review of the influence of phosphate deficiency on the immune response of various plants shows that there is no clear positive or negative direction of the effect [15]. This could indicate that this type of crosstalk between stresses is strongly genotype dependent. Experiments with combined biotic and abiotic stresses considering multiple genotypes are very rare but show an influence of the genotypes used. When combining temperature stress and *Pythium* infection in three *Trifolium* genotypes, You et al. [16] found that the genotypes used also influenced the results. In up to 51 soybean genotypes, Grinnan et al. [17] observed a strong effect of the genotype on the outcome of the combined stresses of drought and insect herbivores. For strawberries, a genotype-dependent effect of inoculum production of *Phytophthora* at different temperatures was also found for the five varieties used [18].

### Black Spot Rose Interaction

Hemibiotrophic *D. rosae* causes widespread black spot disease in roses [19]. In compatible interactions, the fungus forms its first haustoria on subcuticular hyphae within the first 24 h [20]. In addition to the strong growth of hyphae, the first acervuli (reproductive structures) are also formed on detached leaves after approximately five days [21]. In resistant roses, however, *D. rosae* stops growing early. Such incompatible interactions can be recognised microscopically by the fact that, after the penetration of the fungus and the first hyphal growth, no haustoria are formed, and the fungus shows no further development after this short primary growth phase. One gene contributing to resistance against several *D. rosae* races is the TIR-NBS-LRR (Toll/interleukin-1 receptor-nucleotide binding site-leucine rich repeat) (TNL)-type gene *Rdr1* [22]. The rose population 97/7 [23] used here segregates for *Rdr1* and in its resistance status to *D. rosae*. In a previous publication, the expression profiles of the susceptible rose cultivar ‘Pariser Charme’ were revealed using RNA-seq (MACE) [24] and served as the basis for the first expression analyses of *RcPHT* genes under pathogen attack.

The aim of the present study was to investigate the possible relationship between phosphate starvation, resistance to black spot and signalling for phosphate transporters and a defence marker in rose leaves. Furthermore, a strong focus was placed on the effect of genotype on differential gene expression of phosphate transporters and interactions with a hemibiotrophic pathogen.

## 2. Materials and Methods

### 2.1. RNA-Seq (MACE)

The expression changes of the hybrid tea rose cultivar ‘*Pariser Charme*’ (PC) under *D. rosae* and *P. pannosa* infection were described previously (0, 24, 72 h post infection (hpi)) in a MACE/RNA-seq dataset [24]. In addition to the data published there, the hybrid genotype 91/100-5 (91) [25], resistant to *D. rosae*, was also inoculated with this pathogen, and RNA-seq was performed at 24 and 72 hpi. All raw data are available from NCBI under BioProject number PRJNA445241. The procedure was identical to that for PC [24]. Both genotypes were cultivated in a greenhouse under semi controlled conditions, and the detached leaf assay was used for inoculations. The heatmap was created with the help of the R package ‘pheatmap’ [26].

### 2.2. Plant Material

The diploid population 97/7 was originally derived from a backcross of a black spot-resistant cultivar with a susceptible F1 hybrid [23] and is based on a broad spectrum black spot-resistant diploid *R. multiflora* hybrid described earlier [27]. Plants were cultivated in a greenhouse. For the experiment with combined phosphate and pathogen stress, six resistant and six susceptible genotypes of this population were used (s_39, s_69, r_81, s_104, r_123, s_126, r_131, s_183, r_189, r_230, s_239, r_244). From these, a subset of three susceptible (s) and three resistant (r) genotypes were selected (s_104, r_123, s_126, r_131, s_183, r_189) for analysis of gene expression. All 12 genotypes were examined microscopically as described below.

### 2.3. Bioinformatic Methods

Rose PHT genes were identified by BLAST [28] with *A. thaliana* protein sequences of phosphate transporters as queries (Appendix A) [29]. The ‘*Rosa chinensis* Old Blush homozygous genome v2.0′ [30] was used as a reference genome and for gene and protein predictions. Additionally, PHT family sequences of apple were integrated [12].

The alignment of all 121 protein sequences was performed with MEGA X (MEGA X: Molecular Evolutionary Genetics Analysis across computing platforms [31] using MUSCLE (defaulted settings) [32]. The phylogenetic tree was calculated in MEGA X using the maximum likelihood method and JTT matrix-based model [33] with 1000 bootstrap replications. The graphic of the phylogenetic tree was created with iTOL (version 6.3) [34]. The search for transmembrane regions and the corresponding protein families was carried out on the SMART homepage (https://smart.embl.de, accessed on 1 March 2022) [35].

### 2.4. Phosphate Starvation Experiments

Plants will only respond fully to Pi starvation experiments if internal Pi stores have been depleted. Therefore, the mother plants of the 97/7 population were kept under phosphate deficiency until symptoms were visible. This was done in pots in a greenhouse in unfertilized soil. For the nutrition experiments, cuttings were taken from these plants and further cultivated in a hydroponic system.

#### 2.4.1. Pre-Experiment to Determine the Phosphate Starvation-Stress Threshold

To determine the Pi level that allows for sufficient growth with already pronounced phenotypic Pi deficiency symptoms, a pilot experiment was conducted. Cuttings from the prestarved mother plants were rooted for four weeks in tap water and then cultivated in a hydroponic system under different Pi concentrations in the nutrient solution (0, 1, 3, 5, 10 and 100 µM Pi, Appendix A). The hydroponic system consisted of plastic trays (56.8 × 41.8 × 8.6 cm) filled with nutrient solution into which the cuttings were placed via gridded plastic plates holding the cuttings in a fixed grid of nine by seven cuttings. The pilot experiment was conducted in two independent replicates. Leaf length and width (with five technical replicates) and the dry weight shoot/root ratio were determined after eight weeks of culture to define the best possible deficiency situation for our rose genotypes. The threshold value enabled the permanent cultivation of 97/7 mother plants in pots under phosphate stress and maintenance of phosphate deficiency in further hydroponic cultures.

#### 2.4.2. Main Starvation Experiment

The combined nutrition and inoculation experiment (Appendix A) started with taking cuttings from the precultivated and starved mother plants. These were rooted in water for three weeks and then divided between two different Pi nutrient solutions (3 and 100 µM Pi, Appendix A). The nutrient solutions were changed weekly and leaves were harvested for inoculation three weeks after the beginning of the nutrient treatment.

### 2.5. Black Spot Inoculation

The DortE4 isolate of *D. rosae* [36] was propagated on detached PC leaves as described earlier [25]. For the inoculation experiments, freshly unfolded leaves were spray inoculated with a 500,000 conidia/mL suspension or water. The leaves were stored in translucent boxes on moist tissue in an air-conditioned laboratory at 20 °C. Samples were taken and frozen in liquid nitrogen after three days or after 1, 3 and 7 days post infection (dpi) in the time series experiment. In the first experiment, a total of four biological replicates per genotype were tested in two independent inoculation experiments. For the time series, the four biological replicates of genotype s_104 were examined in temporally parallel experiments.

### 2.6. Microscopic Analysis and Determination of Fungal Biomass

For microscopic analysis of fungal development after inoculation, leaf samples of the treatments “D4_fullPi” (*D. rosae* DortE4 inoculation of leaves under full 100 µM P nutrition) and “D4_starvPi” (*D. rosae* DortE4 inoculation of leaves under 3 µM Pi nutrition) were taken at 7 dpi from all six genotypes. Leaves were cut into approximately 1 cm^2^ pieces and were fixed and stained with Alexa Fluor 488-conjugated wheat germ agglutinin (Invitrogen, Carlsbad, CA, USA) as described by Menz et al. [22]. Fluorescence microscopy was performed using a Zeiss Axioscope A1 using filter set 38 HE (Carl Zeiss AG, Oberkochen, Germany).

In addition to checking fungal growth by microscopy, fungal biomass was determined by analyzing fungal and plant ITS regions at the DNA level. For this purpose, 30 mg of leaf material was taken 3 dpi, and the DNA was isolated using a NucleoSpin Plant II kit (Macherey-Nagel, Düren, Germany) according to the manufacturer’s instructions. The amount of plant ITS (ITS-3p53plF1 & ITS-2plR1) [37] and fungal ITS (ITS_Drosae_for, ITS_Drosae_rev) was determined with QuantStudio™ 6 Flex System (Applied Biosystems, Austin, TX, USA) and the Luna^®^ Universal qPCR Master Mix (New England Biolabs, Ipswich, MA, USA) with an annealing temperature of 52 °C. The biomass of the fungus was determined from the ratio of the plant to fungal ITS region. Biomass determination by qPCR was carried out for three resistant (r_123, r_131, r_189) and three susceptible (s_104, s_126, s_183) genotypes.

### 2.7. Analyses of Gene Expression

PCR primers were designed based on the gene- and mRNA predictions for the *R. chinensis* genome [30] using primer3plus [38], and are listed in Appendix A together with their efficiency.

Total RNA was isolated from 30 mg of treated leaf tissue using the Quick-RNA™ MiniPrep Plus kit (Zymo Research, Irvine, CA, USA) according to the manufacturer’s instructions with minor modifications. Dithiothreitol (DTT) was added to the lysis buffer to a final concentration of 50 mM. The leaf tissue was frozen in liquid nitrogen and disrupted using a QIAGEN TissueLyser II (Venlo, The Netherlands). The cDNA was synthesised from 300 ng of total RNA (LunaScript^®^ RT SuperMix Kit, New England Biolabs, Ipswich, MA, USA).

All qPCR analyses and data processing were carried out with the QuantStudio™ 6 Flex System (Applied Biosystems, Austin, TX, USA) and Luna^®^ Universal qPCR Master Mix (New England Biolabs, Ipswich, MA, USA). Two reference genes, SAND and TIP [39], were used for normalisation. Graphs represent the primer efficiency adjusted RQ values (Relative quantification = 2 ^−^^ΔΔCt^) from the QuantStudio™ program. The values for RQ_min_ (RQ_min_ = 2 − (RQ − SE)) and RQ_max_ (RQ_max_ = 2 − (RQ + SE)) were also calculated in this program and are shown as ‘error bars’ in the figures. Expression changes were calculated within genotype for each genotype individually compared with its control. All graphs were created with ggplot2 [40] in R [41]. The test for significance of certain comparisons was performed with REST (V2.0.13, Qiagen, Hilden, Germany; based on [42]), taking into account the primer efficiency. The procedure for the 3′ mRNA-Seq (massive analysis of cDNA ends (MACE)) samples is described by Neu et al. [24].

## 3. Results

### 3.1. PHT Regulation in Inoculated Rose Leaves (RNA-Seq)

The first evidence for special regulation of phosphate transporters in rose was found in a previously published RNA-seq dataset [24]. The susceptible cultivar PC (“PC + D4”) and the resistant genotype 91 (“91 + D4”) were inoculated with *D. rosae*. Expression changes were determined relative to mock inoculated PC leaves. Interestingly, some *PHT* genes were upregulated between 1 dpi and 3 dpi, although the plants in this experiment were not exposed to any phosphate deficiency (Figure 1). The differentially regulated genes were assigned to the following PHT groups by phylogenetic studies: PHT1 (5), PHT2 (1), PHT3 (1), PHT4 (1), PHT5 (1), and PHO (2).

It can be excluded that the upregulation of these genes was due to the uptake of phosphate by the fungus. As in the incompatible interaction of *D. rosae* with 91, fungal growth was stopped before the formation of haustoria (checked using microscopy), preventing the fungus from extracting phosphate from the leaves. However, no differences were observed when comparing the susceptible (“PC + D4”) and resistant (“91 + D4”) genotypes. In the incompatible variant (“91 + D4”), the same genes were regulated to a similar extent as in the compatible interaction. The only exception is the slight downregulation of the RC4G0373800 (*RcPHT2-1*) and RC4G0403200 (*RcPHO-H1*) genes and the slight induction of RC5G0591700 (*RcPHT4-1*), which is only found in 91 + D4 and not in PC + D4.

Overall, we observed that members of the *PHT1* family were predominantly induced. Therefore, we focused on selected members of the *PHT1* family, particularly *RcPHT1-9* and *RcPHT1-4*. However, for the study of this dataset, the complex family of phosphate transporters in roses had to be investigated in more detail.

### 3.2. Bioinformatic Analysis of the PHT Family in Roses

#### 3.2.1. Screening of One *R. chinensis* Genome Revealed 38 RcPHT Genes

Orthologue rose *PHT* genes were identified by BLAST with 27 *A. thaliana* protein sequences of phosphate transporters (Appendix A) against the gene models of the *R. chinensis* genome [30]. A total of 38 potential rose phosphate transporters were identified. The rose *PHT* genes are distributed throughout the genome and to not show a prominent clustering on one chromosome. Only chromosome 3 is very poorly represented, with only one gene (RC3G0282600).

For the *PHT1* family, 11 gene predictions for roses were found, the predicted number of transmembrane regions (TMRs) was 10 to 12, and the proteins all belong to the major facilitator, sugar transporter-like protein family (InterPro: IPR036259). RC7G0352000 and RC7G0352200 are also predicted to belong to the phosphate permease subgroup (InterPro: IPR004738). The single PHT2 protein RC4G0373800 is predicted to have 11 TMRs and to belong to the phosphate transporter family (IPR001204). Three rose sequences were found for the PHT3 family, which are assigned to the protein families mitochondrial carrier domain superfamily (IPR023395), and mitochondrial phosphate carrier protein Pic2/Mir1-like (IPR044677) and do not contain TMRs. Seven rose candidates for PHT4 were found. The proteins were predicted to have 9 to 12 TMRs and belong to the major facilitator superfamily (IPR011701) and solute carrier family 17 member 9-like (IPR044777) protein families. Another seven rose sequences have been identified as members of the PHT5 family. Of these, five are attributed to the SPX domain-containing protein (IPR031142) and do not appear to have TMRs. The proteins RC1G0313400 and RC7G0208500 are attributed to the major facilitator superfamily (IPR011701) and were predicted to have 9 and 11 TMRs, respectively. Nine potential members of the PHO family were found. No affiliation with a protein family was found. For the *Arabidopsis* proteins used, the InterPro scan did not reveal any specific affiliation using SMART. Except for RC4G0427700 and RC4G0427800 without TMRs, three to eight TMRs were predicted.

#### 3.2.2. Phylogenetic Analyses of Rose PHT like Sequences

Sequences of *Arabidopsis* PHT proteins were aligned with the corresponding rose genes (Appendix A), and a phylogenetic tree was computed together with PHT proteins characterised from apple [12]. The division of the protein sequences into the different PHT families is reflected by clearly separated clusters within the tree (Figure 2).

Cluster “PHT1” groups together 10 *At*, 11 *Rc* and 16 *Md* protein sequences. A clear assignment of each rose sequence to an *Arabidopsis* sequence was not possible. The proteins RC7G0352000 and RC7G0352200 can be assigned to the AtPHT1-9 and AtPHT1-8 groups. These proteins also split off very early from the other sequences of this cluster. The corresponding apple sequence is called MdPHT1-4. RC6G0380800 and RC6G0380900 cluster with AtPHT1-4 and AtPHT1-7. The other sequences show greater similarities to the apple proteins. The small group of PHT2s consists of one *Arabidopsis* and one *Rosa* sequence (RC4G0373800) and two apple proteins. More *Malus* proteins (nine) were also found for the PHT3 cluster than for *Arabidopsis* and *Rosa* (three each). The assignment and naming of these proteins is clear in this case, RC2G0689300 as RcPHT3-1, RC7G0112800 as RcPHT3-2 and RC6G0314600 as RcPHT3-3. The cluster PHT4 comprises eight *At*, seven *Rc* and 12 *Md* proteins. The assignment of the rose sequences is as follows: RC5G0591700 (RcPHT4-1), RC3G0282600 (RcPHT4-2), RC5G0588100 (RcPHT4-3), RC1G0348100 (RcPHT4-4), RC2G0027200 (RcPHT4-5) and two sequences for RcPHT4-6 (RC5G0444000, RC6G0313900). The names of the apple proteins do not match those of *Arabidopsis*.

The cluster of SPX-containing proteins (PHT5) contains seven *At* and *Rc* proteins each and four apple proteins. The assignment to the *At*SPX proteins was unambiguous (RC2G0037600 (RcSPX1), RC6G0464900 (RcSPX2), RC1G0286700 and RC1G0286800 (RcSPX3) and RC2G0615400 (RcSPX4)). However, the rose sequences RC1G0313400 and RC7G0208500 show a greater proximity to the MdPHT5 sequences.

The cluster of PHO proteins consists of nine *Arabidopsis* and 11 rose proteins, and no apple sequences were included. Clear allocations were made for RC4G0058000 and RC4G0058200 (RcPHO1), RC4G0403200 (RcPHO-H1) and RC2G0615200 (RcPHO-H9). RC4G0427600 clusters with most of the remaining *Arabidopsis* sequences (AtPHO homologues 2 to 7).

### 3.3. Determination of the Stress Level for the Pi Starvation Experiments

A total of six resistant and six susceptible genotypes of population 97/7 [23] were chosen for the Pi starvation experiments. Different growth parameters were measured at various Pi levels (Figure 3). Leaf size was significantly smaller at Pi concentrations of 0 µM to 10 µM compared to fully nourished roses (100 µM). This effect was also observed for the shoot/root ratio.

A level of 3 µM Pi was found to be optimal for stress treatment, as there were clear phosphate deficiency symptoms, but the leaf sizes achieved were still sufficient for fungal inoculation experiments. To keep the Pi deficiency as high as possible, 3 µM rather than 5 µM Pi was chosen, although both gave comparable results. This Pi fertilizer level was used for both the nutrition experiments and the permanent cultivation of the mother plants.

### 3.4. Effect of Pi Starvation on Black Spot Infection (Microscopy and Biomass)

To analyse the putative interactions between black spot infection and Pi starvation, two fully independent inoculation trials in two different years, each with two biological replicates, were conducted under full Pi and starved Pi conditions. This was done to account for the frequently observed variability between independent inoculation experiments, and is ignored in most published reports on plant pathogen interactions. The trails used six resistant and six susceptible genotypes.

To analyse the interactions in more detail, samples were taken at seven dpi for microscopic analysis of fungal structures (Figure 4). In the resistant genotypes, a stop in growth could be observed before the first haustoria were formed. The phosphate status did not change the morphology of the incompatible interaction. On the other hand, there was strong fungal growth on the leaves of the susceptible genotypes accompanied by the formation of haustoria and later formation of acervuli with new spores. Here, it could not be determined that a phosphate deficiency had an effect on the development of *D. rosae*.

In addition to the microscopic analyses, fungal biomass was determined by qPCR of DNA extracted from inoculated and control leaves. This analysis also showed no significant evidence of a change in fungal biomass due to the different phosphate treatments (Appendix A). However, as expected, the biomass of *D. rosae* differed strongly depending on the resistance status of the genotype.

### 3.5. Effect of Pi Starvation and Black Spot on the Gene Expression of RcPHT Genes and the RcPR1 Defence Gene

To analyse the effect of Pi starvation and black spot infection on *RcPHT1-9* (RC7G0352200) and *RcPHT1-4* (RC6G0380900) expression, we analysed gene expression at 3 dpi, as the differences between compatible and incompatible interactions were already fully developed [24]. As a marker for the defence response, the gene *RcPR1* was included. The relative changes in expression (using RT-qPCR) compared to fully nourished and mock-inoculated leaves (“H_2_O_fullPi”) are shown in Figure 5. In this first analysis, the expression data of all six genotypes were combined.

Here, no significant changes in the expression of the phosphate transporters *RcPHT1-4* and *RcPHT1-9* due to inoculation with *D. rosae* alone could be observed. In contrast, both genes were upregulated by Pi starvation. *RcPHT1-9* was upregulated in Pi-starved leaves compared to the fully nourished control (“H_2_O_starvPi” vs. “H_2_O_fullPi”) by a factor of 13.013 (*p* < 0.000). For the gene *RcPHT1-4,* a slight upregulation by a factor of 2.178 (*p* = 0.000) was detected.

The gene *RcPR1*, which is a general marker for plant-pathogen interactions that was previously shown to be induced by black spot infection [24], was analysed as well. It displayed a slightly higher, although statistically nonsignificant, induction when black spot inoculation was combined with Pi starvation.

### 3.6. Differences in Gene Expression between Pathogen Interaction Types

To investigate a possible effect of the resistance status of the host plant, the expression changes were also calculated separately for groups of resistant and susceptible genotypes. Figure 6 shows the relative expression changes, and Appendix A provides further statistical data. Compatible interactions seem to have stronger expression values for most genes after inoculation with black spot. However, this was weakly significant only for *RcPTH1-4* (*p* = 0.025, Appendix A). In contrast to the RNA-seq results, *RcPHT1-9* was only induced by Pi deficiency and not by black spot inoculation. *RcPR1* was only induced upon pathogen inoculation and not by Pi starvation, with slight but statistically nonsignificant differences between compatible and incompatible interactions.

Since the separate analyses of susceptible and resistant genotypes indicated a tendency towards higher expression values under combined stress expression, the patterns for each individual genotype were examined in more detail. Here, significant differences were observed for all three genes investigated (Figure 7).

### 3.7. Differences in Gene Expression in Individual Genotypes

The effects of gene expression in relation to phosphate starvation and interaction with black spot displayed strong genotype dependent effects independent of the interaction type (compatible or incompatible, Figure 7, Appendix A). Figure 7 shows the differences between the genotypes within each treatment, and Appendix A shows statistics on comparisons between treatments for each genotype separately.

A few examples of each treatment demonstrate these genotype-specific expression patterns (Figure 7 and Appendix A):Upregulation of *PHT* genes due to Pi starvation only (“H_2_O_fullPi” vs. “H_2_O_starvPi”) is genotype-specific in both the extent and significance. As an example, significant upregulation was observed in genotype r_123 for both *RcPHT1-4* (3.1x ×, *p* = 0.001) and *RcPHT1-9* (31.2×, *p* = 0), whereas it was not significant in genotype s_126.The regulation of *PHT* genes by black spot inoculation alone (“H_2_O_fullPi” vs. “D4_fullPi”) was significant for only some genotypes, and only to a minor extent. *RcPHT1-9* was slightly upregulated in genotype s_104 and slightly downregulated in s_183. As we observed profuse development in fungal biomass in all susceptible genotypes, the lack of *PHT* induction in the latter case indicates that there is no strict correlation of *PHT* upregulation with susceptibility.Synergistic effects of combined stress factors (black spot interaction and starvation) were observed in only a subset of genotypes for the two phosphate transporters (“H_2_0_starvPi” vs. “H_2_O_fullPi”). As an example, *RcPHT1-9* was significantly upregulated only in genotype s_126 (3.2×, *p* = 0.02) and, as a trend, also in r_183 (14.6×, but with only two replicates in “H_2_O_starvPi”), whereas it was not significantly downregulated (r_123, s_183) or upregulated (r_131, s_104) in other genotypes. *RcPHT1-4* showed no significant changes, with at least two-fold changes. For *RcPR1*, a significant upregulation of combined stressors (“D4_fullPi” vs. “D4_starvPi”) was only observed in genotypes r_189 (3.2×, *p* = 0.006) and s_104 (12,9×, *p* = 0.001). Here, again, we see an example of how resistance status does not seem to play a role in these different expression patterns.Even the *RcPR1* pathogen interaction marker, which was significantly upregulated in all genotypes after black spot inoculation (“H_2_O_fullPi” vs. “D4_fullPi”), showed strong and significant differences in the degree of upregulation among genotypes (Figure 7).

In summary, the expression of all three genes varied greatly between different genotypes in the single and combined stress factor experiments, although they were full sibs from the same cross. The combined stress factors Pi starvation and black spot interaction had a significant effect on gene expression in some genotypes.

### 3.8. Time Series Experiment with Genotype S_104

To further elucidate the reproducibility, a third independent experiment with a time series was conducted only for the genotype s_104 with sampling at one, three and seven dpi, and the same treatments already described. The expression of *RcPHT1-9* and *RcPR1* relative to the control (“H_2_O_fullPi”) is shown in Figure 8.

In this experiment, *RcPR1* was only upregulated upon infection with black spot, but not upon Pi starvation, which is consistent with the results of the previous two experiments. Significant differences were also found between “D4_fullPi” and “D4_starvPi”. Because of only two biological replicates, the difference at one dpi is not statistically significant, but at three dpi (3.1×, *p* = 0.037).

For the phosphate transporter *RcPHT1-9*, induction by the Pi starvation treatment without additional black spot inoculation was seen at one dpi but not at three and seven dpi, which is in contrast to the previous two experiments where we observed strong induction at three dpi. However, at all time points, a strong induction was observed in the combination of black spot inoculation and Pi starvation (“D4_starvPi”), which was also always higher than in “H2O_starvPi” (1 dpi: 4.4×, *p* = 0.014; 3 dpi: 5×, *p* = 0.001; seven dpi: 4.4×, *p* = 0). This latter observation is in agreement with the previous two experiments where a similar tendency was seen (Figure 7). This is an indication of the synergistic effects of the two stressors.

Despite the standardised fungal propagation and inoculation and the treatment of mother plants and cuttings, it was not possible to achieve completely identical results at different times. Once again, the experimental results illustrate the need to include several genotypes in experiments and to carry out experiments using independent replicates.

## 4. Discussion

The results presented here demonstrate that stress experiments involving plant pathogen interactions and/or a combination of biotic and abiotic stress factors might display extreme genotypic effects that remain undetected if only single genotypes or a combination of two genotypes with contrasting phenotypes (resistant vs. susceptible, starvation tolerant vs. starvation intolerant) are studied.

Our rationale to study the expression dynamics of two *RcPHT1* genes is based on past MACE analyses of inoculation experiments in rose leaves conducted with one resistant (91/100-5) and one susceptible (PC) rose genotype (partly published in [24]). These genes were selected for the present analyses, as their induction by plant pathogen interactions was supported by the in silico analysis of Baek et al. [14], who described possible biotic stress-related promoter binding sites for many *AtPHT*s. However, the direct influence of phosphate deficiency on the expression of *RcPHT*s and other pathogen-related genes was not addressed in that publication. We analysed three resistant and three susceptible genotypes of a segregating population of diploid roses that only share the presence or absence of a single dominant resistance gene (Rdr1) against the fungal isolate used in the inoculated samples (the DortE4 isolate here abbreviated as “D4”) [22]. Therefore, completely unrelated individuals not belonging to a full sib family might even display larger variation; those differences might go undetected in pooled samples where different genotypes are combined under different stress treatments. The results support the view that a reduction to a single genotype would not have provided transferable data, as major differences between the genotypes became visible. Other studies with combined stress also came to this conclusion [16,17,18].

Our efforts to include natural variation in stress treatment experiments go even further; we performed our main experiment in two completely independent experiments in 2015 and 2016 comprising two completely independent stress induction experiments as well as two completely independent fungal inoculation experiments with two biological replicates each. A third such experiment was conducted in 2020 for the time course experiment on a single genotype. This is in contrast to most published experiments where a single treatment is conducted and time points and biological experiments are defined at best as individual repeats of clonal plants or plant lines in the experiments. Therefore, we observed a significant variation in the repeat samples of our expression data, which created significant background noise, but it might also partially exclude random effects that less complex experiments could create.

Irrespective of the large variation in our data, we observed some effects and trends of both black spot colonization and Pi starvation, as well as the combination.

### 4.1. The Putative Phosphate Transporters RcPHT1-9 and RcPHT1-4 Are Upregulated in Leaves If Plants Are Cultivated under Low Phosphate Conditions

Upregulation was much stronger in *RcPHT1-9* compared to a somewhat lower regulation of *RcPHT1-4,* but this was strongly genotype dependent in both cases. In the literature, their expression in apple and *Arabidopsis* has also been described differently. In apple, the orthologue to *RcPHT1-9* is named *MdPHT1-4*, and its expression was slightly downregulated by phosphate deficiency in leaves but induced after the combined stress of deficiency and drought. Deficiency of phosphate alone did not lead to upregulation, even in roots [12]. Conversely, [43] described *AtPHT1-4* as particularly strongly induced under phosphate deficiency. In their experiments, *AtPHT1-9* showed only slight activation. However, the authors examined the expression of roots. Additionally, Mudge et al. [10] observed only slightly increased expression of *AtPHT1-9* in *Arabidopsis* roots. However, Remy et al. [44] experiments detected a strong upregulation of *AtPHT1-9* in roots. Furthermore, leaves under phosphate deficiency did not show any *AtPHT1-9* upregulation, only senescent leaves showed a strong induction, and an upregulation of *AtPHT1-4* was observed in Pi-starved shoots.

### 4.2. Black Spot Disease without Pi Starvation Leads to Expression Changes of RcPHT1-9 Only in Two Genotypes

In the 97/7 population, upregulation was only observed in genotype s_104 (2,8×, *p* = 0.037) and downregulation was observed in s_183 (0.04, *p* = 0) due to interaction with *D. rosae* with sufficient Pi supply (“H_2_O_fullPi” vs. “D4_fullPi”). Although in contrast with our previous analyses of MACE data [24], this observation demonstrates some interaction between pathogen infection and the regulation of the phosphate transporter *RcPHT1-9*. Nevertheless, we did not see the same extent of induction as in the MACE experiment. Apart from the genotypic effect that we demonstrate in the present study, another explanation may be the differences in the analysis methodology. A variation of the RNA-seq method was used in one case and RT-qPCR was applied in the current work. Analyses of gene expression show numerous examples of differences in the extent of gene expression if measured with different technologies, especially for smaller and lower expressed genes (e.g., [45]).

### 4.3. Interactions of Stress Factors in the Induction of RcPHT1-9 and RcPHT1-4

The regulation of *RcPHT1-9* and *RcPHT1-4* by both Pi starvation and black spot infection is also apparent from another observation. We observed a change in expression when two stressors were combined compared to treatment with only one stressor. It is possible that for some genotypes, this combination causes minor, statistically nonsignificant effects to be more pronounced when the stressors act in combination than with only one stressor. An example is genotype s_126, where *RcPHT1-9* was not upregulated after only black spot infection (0.94×, *p* = 0.8) or Pi starvation alone (2.347×, *p* = 0.055), but was significantly upregulated when both stresses were combined (3.2 over Pi-starvation alone, *p* = 0.02).

An increase in induction by the combination of stressors in our system is also supported by an increased induction of the pathogen interaction marker *RcPR1*. This shows a higher induction under combined stress compared to induction by black spot inoculation alone in genotypes r_189 and s_104 (“D4_fullPi” vs. “D4_starvPi”). This indicates a signalling crosstalk between the stress response to phosphate starvation and the response to pathogens in a genotype-specific manner. Examples of this can be found in the literature, e.g., Tang et al. [46], Castrillo et al. [47] and Dong et al. [48] observed a negative influence of Pi Stress on the immune response, Khan et al. [49] detected a positive effect, and Campos-Soriano et al. [50] observed a negative influence with an excess of Pi.

### 4.4. Pi Starvation Does Not Change the Rose Black Spot Interaction

Our microscopy data showing compatible and incompatible interactions as well as the quantification of fungal biomass via ITS-PCR do not show a switch of compatibility under Pi starvation. Furthermore, no changes in the relative abundance of the fungus were found. This indicates that, although it affects the gene expression of pathogen-responsive genes in some genotypes, Pi starvation does not significantly affect the colonization with *D. rosae* in roses. However, given the variation we observed in gene expression among genotypes, this might be different in other genetic backgrounds. In many host-pathogen interactions, however, an influence of Pi on disease severity could be observed, although contradictory results were also obtained [15].

## 5. Conclusions

In the present study, we showed that Pi starvation of whole rose plants and black spot infection of rose leaves both affect the expression of phosphate transporter genes and that the two stressors interact with each other. However, these effects are strongly genotype dependent and varied significantly in their extent between completely independent experiments. We conclude that many biological processes cannot be studied in a meaningful way in single genotypes but rather must be analysed based on experiments with a broader set of genotypes within each species. In this work, the family of rose phosphate transporters was studied in detail, and the results provide a good starting point for further studies on phosphate deficiency.

## Figures and Tables

**Figure 1 jof-08-00549-f001:**
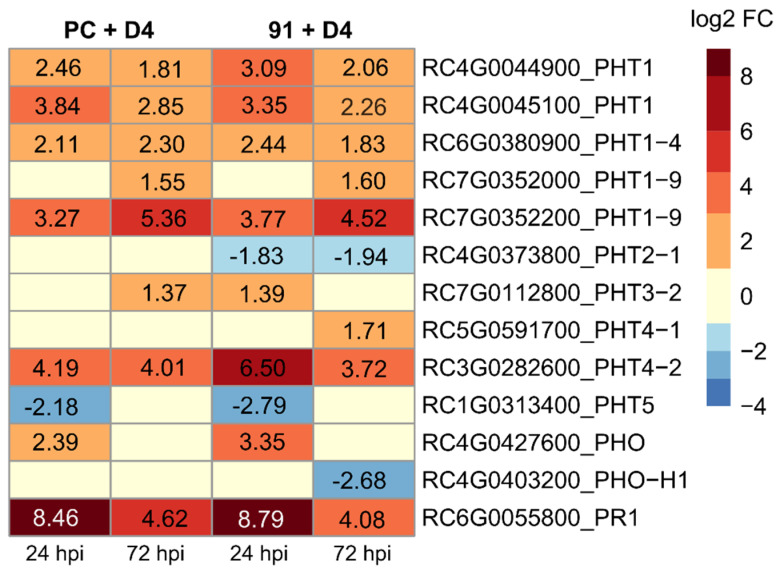
Heatmap showing transcriptional changes relative to mock inoculation of several phosphate transporters in rose leaves after inoculation with the fungal pathogen *D. rosae* (“D4”) at 24 and 72 hpi. PC: susceptible ‘Pariser Charme’, 91: resistant to “D4”. Additionally the regulation of RcPR1 is shown. Only RcPHTs with significant changes are included, *n* = 3.

**Figure 2 jof-08-00549-f002:**
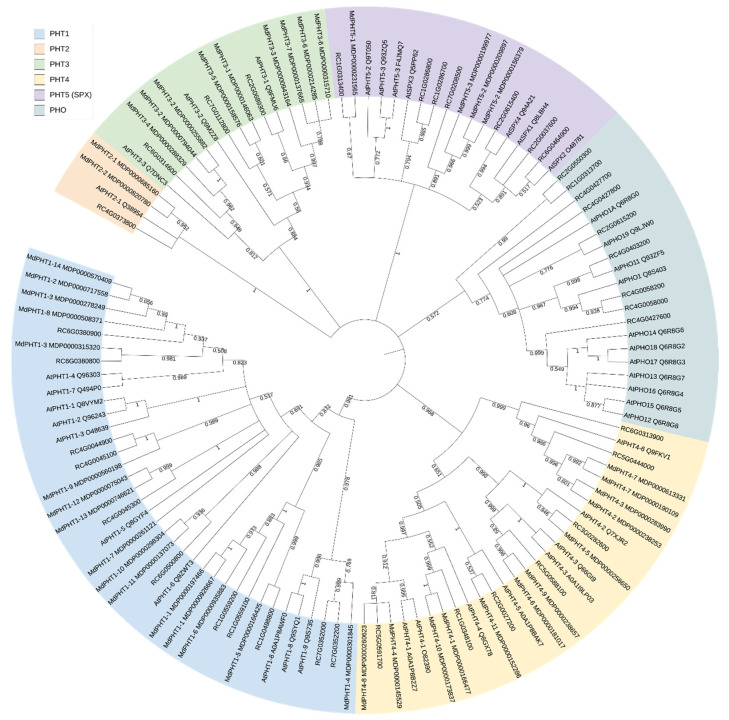
Phylogenetic analyses of PHT proteins in *Arabidopsis thaliana* (At), *Rosa chinensis* (Rc) and *Malus domestica* (Md). Evolutionary analysis by maximum likelihood method and JTT matrix-based model using MEGA X and 1000 bootstraps. The percentage of trees in which the associated taxa clustered together is shown next to the branches.

**Figure 3 jof-08-00549-f003:**
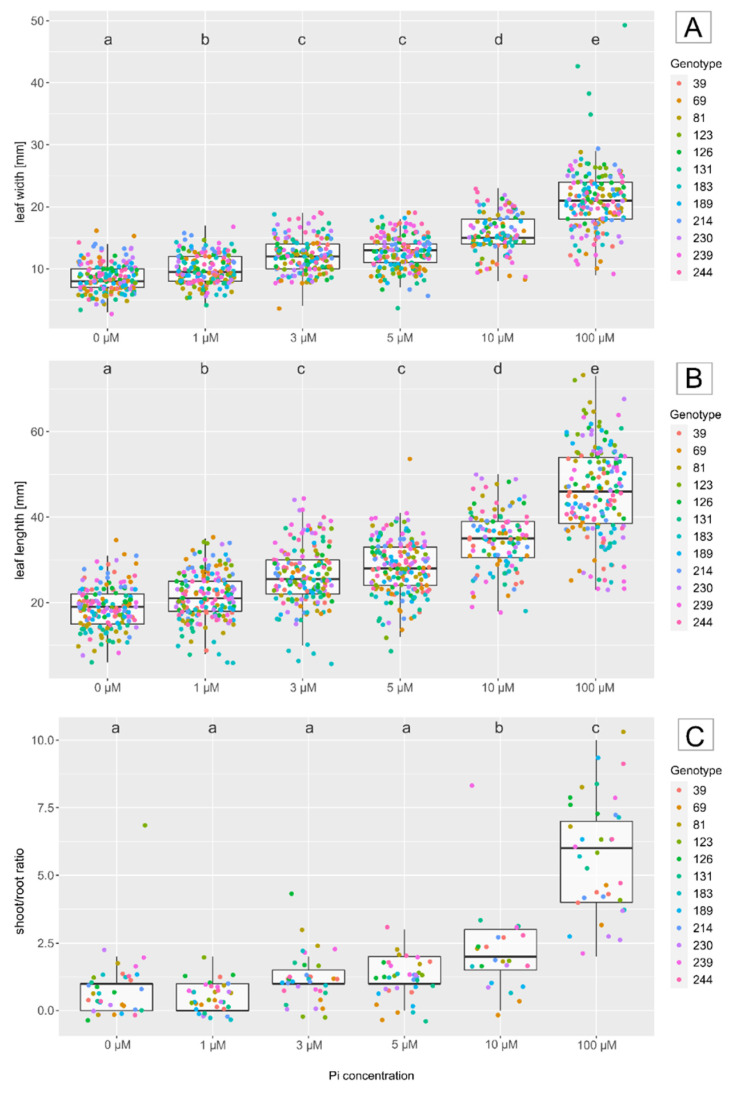
Boxplot of growth parameters of leaves. The leaf width (mm) (**A**), leaf length (mm) (**B**) and dry weight shoot/root ratio (**C**) under different P concentrations (µM) in the nutrient solution. Letters indicate significant differences between all treatments (Tukey test, Bonferroni adjusted).

**Figure 4 jof-08-00549-f004:**
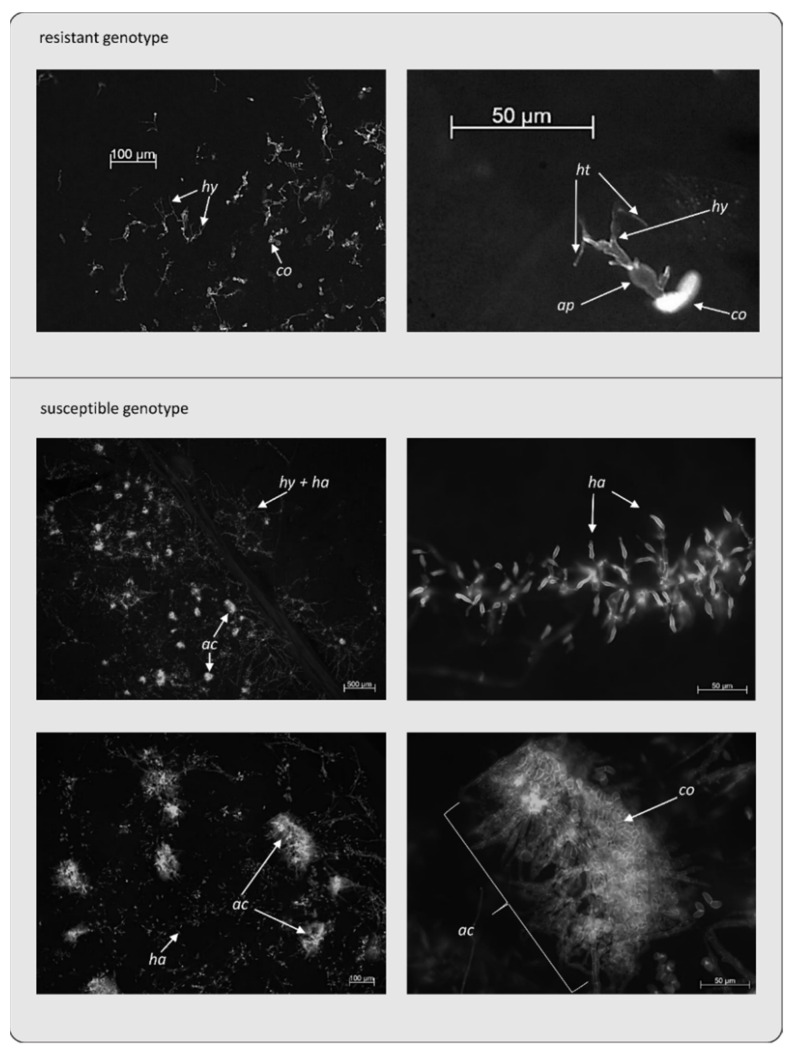
Microscopic analysis of resistant and susceptible 97/7 genotypes. Example of *D. rosae* growth on leaves of the resistant r_244 and susceptible s_104 genotypes at 7 dpi. The arrows point to co: conidium, ap: appressorium, hy: hyphae, ha: haustorium, ht: haustorial tube, ac: acervulus. Samples were stained with Alexa Fluor 488-conjugated wheat germ agglutinin (WGA).

**Figure 5 jof-08-00549-f005:**
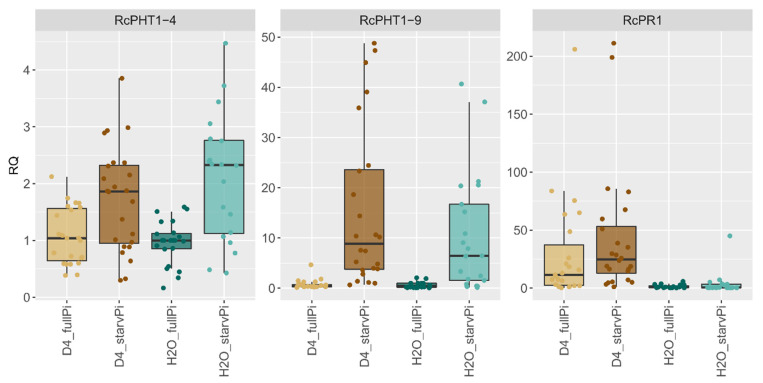
RT-qPCR analysis of a combined stress experiment in leaves of six 97/7 rose genotypes at 3 dpi. Two Pi treatments (“fullPi” and “starvPi”) and additional inoculation with *D. rosae* (“D4”). Expression data of all six genotypes have been combined under each treatment. Expression changes of each genotype relative to “H_2_O_fullPi”. Data are represented as boxplots of RQ values (relative quantification = 2 ^−^^ΔΔCt^) and single points, *n* = 24.

**Figure 6 jof-08-00549-f006:**
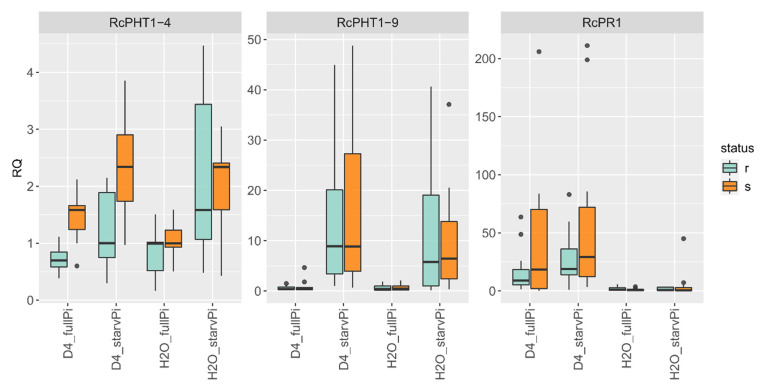
Expression changes of groups of resistant (r_123, r_131, r_189) or susceptible (s_104, s_126, s_183) genotypes. RT-qPCR analysis of a combined stress experiment in leaves of six 97/7 rose genotypes at 3 dpi. Two Pi treatments (“fullPi” and “starvPi”) and additional inoculation with *D. rosae* (“D4”). Expression changes of each genotype relative to “H_2_O_fullPi”. Data are represented as boxplots of RQ values (relative quantification = 2^−^^ΔΔCt^), *n* = 12. Appendix A provides further data.

**Figure 7 jof-08-00549-f007:**
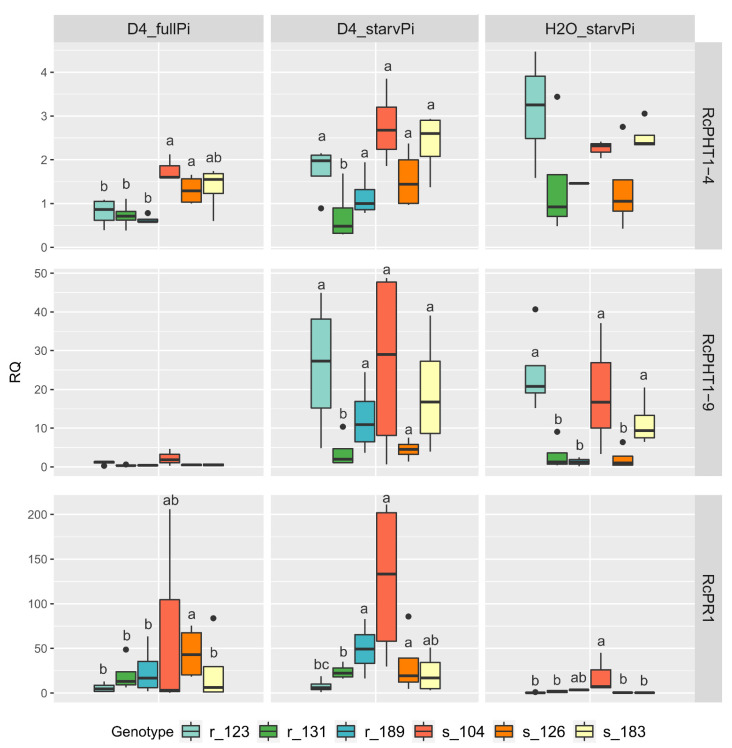
RT-qPCR analysis of a combined stress experiment in leaves of six 97/7 rose genotypes at 3 dpi. Two Pi treatments (“fullPi” and “starvPi”) and additional inoculation with *D. rosae* (“D4”). Expression changes of each genotype relative to “H_2_O_fullPi”. Data are represented as boxplots of RQ values (relative quantification = 2^−^^ΔΔCt^). The letters indicate groups of genotypes whose gene expressions differed significantly within a treatment (REST). The expression changes caused by the different treatments for each genotype are shown in Appendix A. r: resistant, s: susceptible genotype, *n* = 4.

**Figure 8 jof-08-00549-f008:**
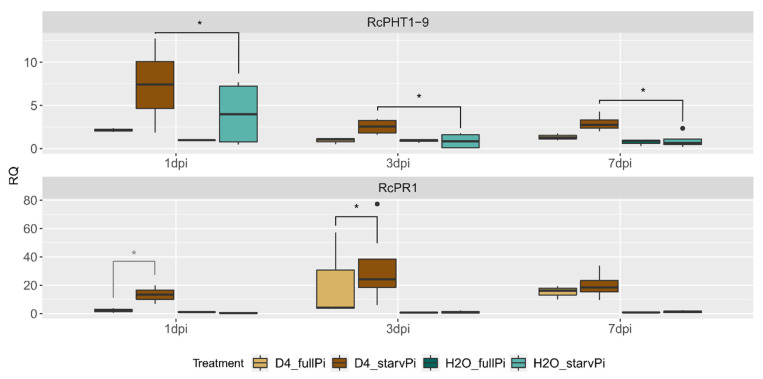
RT-qPCR analysis of a combined stress experiment in leaves of rose genotype s_104 at one, three, and seven dpi. Two Pi treatments (“fullPi” and “starvPi”) and additional inoculation with *D. rosae* (“D4”). Expression changes relative to “H_2_O_fullPi” at each time point. Data are represented as boxplots of RQ values (relative quantification = 2^−^^ΔΔCt^) and some significant comparisons are marked by asterisks (in grey because of *n* = 2), *n* = 4.

## Data Availability

qPCR raw data are available from NCBI under BioProject number PRJNA445241.

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
