# Peer review of "P Starvation in Roses Leads to Strongly Genotype-Dependent Induction of P-Transporter Genes during Black Spot Leaf Disease"

_jof, 2022, doi:10.3390/jof8060549_

Round 1

Reviewer 1 Report

The article is well written. There are issues to be reviewed by the authors.

  1. the introduction part is too long. It's a kind of review of the literature. Needed to write again. Especially, paragraph 3 & 4 is not necessary (L71-80 and L81-96).  L99-112: this part could be reduced. There are differences in the discussion and introduction.

Author Response

Thank you very much for your helpful comments.

Although one of the other reviewers positively evaluates the introduction we agree that it is a bit long. Therefore, we follow some of your suggestions in deleting some of the sections. However, we think that contradictory results on the effect of Pi on resistance reactions are crucial to mention in the introduction and therefore kept this section as well as a short introduction on black spot as this might be unfamiliar to most of the readers of Jof.

Reviewer 2 Report

The manuscript by Domes et al. deals with elucidating plant-microbe interaction focusing on black spot infection, phosphate transporters expression, and one defense marker gene in roses. They evaluate the expression of phosphate (Pi) transporters upon Pi starvation in different genotypes. They claim Pi starvation can induce several phosphate transporter genes during pathogen interactions. They also observed that the marker gene RcPR1 expression is induced by black spot infection in a genotype-dependent manner. 

I appreciate the efforts of the authors, but overall, the presented data are insufficient, statistically unexpressive; thus, the results do not support the authors' claim. 

Author Response

Dear reviewer,

we are sorry that our manuscript does not find your approval. As we understand your comments you vote for rejecting it. Unfortunately, without any detailed suggestions for revisions we can not provide any revisions.

Reviewer 3 Report

The manuscript describes a study of black spot infection and leaf expression of phosphate transporters in roses under P starvation with the goal to investigate the effect of P starvation on gene expression of phosphate transporters. The basic conclusion is that P starvation, either whole rose plant or black spot infection, affects phosphate transporter expression. Generally, I found this manuscript well written. The introduction leads to a clear hypothesis. The methods are appropriate to deal with these hypotheses, and the results are clearly presented. I appreciate their good piece of work.

Author Response

Dear reviewer,

we are very pleased that our manuscript finds your approval. As you see from our comments to reviewer 1 and 4 we tried to improve our introduction, materials and methods and some parts of the results.

Reviewer 4 Report

This study addresses an important question in fungal-plant interactions: the synergism between disease development and plant gene expression related to defense and stress response, in the framework of the plant genotype. Specifically, the phosphate transporter genes of rose were identified and three factors considered:  phosphate starvation versus phosphate-replete conditions; black spot infection vs. mock inoculation and different susceptible and resistant progeny of a biparental rose population. The results are interesting. Some technical issues would need to be addressed:

major:

The statistical basis for the conclusions seems to develop in waves over the datasets presented in the different figures. To strengthen the conclusions of the paper, could it be possible to extract all data belonging to the same treatment, normalize to permit comparison of experiments done on different days, and replot?

The statistical comparisons seem to be based on a proprietary software, REST, please explain what tests are used by the system.

Figure S2 - where is the blue box? it seems that the "all" category is not shown in the figure? are the error bars SE or SD, and why is the D4_fullPi, sensitive strain, assymetric for the up/down is RQ is linear, correct?

Figure 5 six genotypes - I see only one genotype, three transporter genes, 4 treatments. OK so in the text it says that all six genotypes were combined. Is this cause of the big variability or is the variability random with across genotypes? (this relates to my main comment, above)

some other notes, to address during revision:

Figure 2 - what are the dashed lines?

line 45 - enlarge - better enhance; also "increasing numbers" better "increasing abundance" or a similar term

line 91 - this is sort of general remark and could be deleted

line 233 - not yet = not previously published? are they published here? in the supplement? deposited in a database?

line 282 - why "no affiliation" if they are members of the PHO family?

line 343 typo trails

In the abstract, make clear that PHT are plant genes (because this is a fungal journal).

Author Response

major:

The statistical basis for the conclusions seems to develop in waves over the datasets presented in the different figures. To strengthen the conclusions of the paper, could it be possible to extract all data belonging to the same treatment, normalize to permit comparison of experiments done on different days, and replot?

Response:

  • The first dataset representing Pi starvation is represented by three figures: 5, 6 and 7. Based on the same original dataset the presentation is meant to focus on three levels of complexity with figure 5 combining all genotypes within one treatment, figure 6 separating compatible and incompatible interactions with three genotypes combined in each category and figure 7 resolving the differences between each genotype. We do not see any way to unify this part of the data, as it is important for our point that genotypic differences at different levels (compatible, vs. incompatible or even individual genotypes) do have an effect on stress response experiments. Figure 8 represents a third inoculation experiment with only one genotype; however, with more time points so that here as well we think an integration into the first dataset is difficult. More so because here we also see variation between independent experiments conducted a year later than the other experiments. However, as this addresses the problem of reproducibility of research experiments in more or less different environments we would like to keep this separation to make our point on this issue.

The statistical comparisons seem to be based on a proprietary software, REST, please explain what tests are used by the system.

Response:

  • The program REST (V2.0.13, Qiagen, Hilden, Germany) is based on a work of Pfaffl et al. (2002; doi: 1093/nar/30.9.e36) and is using a Pair Wise Fixed Reallocation Randomisation Test. We added this to the material&method section.

Figure S2 - where is the blue box? it seems that the "all" category is not shown in the figure? are the error bars SE or SD, and why is the D4_fullPi, sensitive strain, assymetric for the up/down is RQ is linear, correct?

Response:

  • In Fig S2, data from all six genotypes were summed for mock inoculation without distinction between resistant or susceptible. For the relative expression change, these data were set to 1, which is why they are almost invisible in the plot. The "error bars" in all figures represent RQmin (RQmin = 2 - (RQ - SE)) and RQmax (RQmax = 2 - (RQ + SE)) (SE=standard error). We have included this in the Material&Methods section.

Figure 5 six genotypes - I see only one genotype, three transporter genes, 4 treatments. OK so in the text it says that all six genotypes were combined. Is this cause of the big variability or is the variability random with across genotypes? (this relates to my main comment, above)

Response:

  • In figure 5 all six genotypes have been combined under each treatment listed at the x-axis. The large variability is due to the genotypic differences but serves to display overall reaction patterns (e.g. that PHT1-9 is induce only under Pi starvation or that PR1 is only induced upon contact to pathogens but stronger if stressors are combined). We added a sentence to the figure description to make this clearer.

some other notes, to address during revision:

Figure 2 - what are the dashed lines?

Response:

  • Figure 2: The dashed lines are an artifact of the formatting process done by JoF. Our original figure does not contain dashed lines. We will contact the editorial office to point out this problem.

line 45 - enlarge - better enhance; also "increasing numbers" better "increasing abundance" or a similar term

Response:

  • We changed “enlarge” to “enhance” and “number” to “abundance”

line 91 - this is sort of general remark and could be deleted

Response:

  • We agree and deleted the sentence

line 233 - not yet = not previously published? are they published here? in the supplement? deposited in a database?

Response:

  • The raw RNAseq data will be publicly available (NCBI BioProject: PRJNA445241), but genotype 91 data have not yet been analyzed for publication. Because of the misunderstanding, we have removed the sentence and adjusted materials&methods section.

line 282 - why "no affiliation" if they are members of the PHO family?

Response:

  • The tool (https://smart.embl.de ) we used did not find any specific affiliation. We have not used any other software tool.

line 343 typo trails

Response:

  • We have corrected this

In the abstract, make clear that PHT are plant genes (because this is a fungal journal).

 Response:

  • The abstract was edited to explain that PHT are plant genes.

Round 2

Reviewer 4 Report

The authors' reply to my major comment on the first version is quite convincing, and I fully agree with the aim to highlight the differences between independent data sets.

Nevertheless my opinion remains that some (not all) datasets might be combined. Specifically, the dark-orange bars in Figure 7, line s_104, look clearly different from at least some of the sensitive strains. The genetic basis for this is not the same as compatible vs incompatible, and it will certainly not be possible to identify the genes responsible at this stage. At the descriptive level, though, perhaps normalizing to a high qPCR value rather than a low one (fullPi-H2O) would help? For example, in the top row, second panel of Figure 7, s_104 and r_189 show a large difference but it is not statistically significant. Furthermore it looks like there is some technical problem with normalization because the average fullPi-H2O values after normalization are not always equal to 1 (for example panel three from the left, top row, Figure 7). 

Therefore, I would recommend to look once more at my comment on the first version, in the hope that the conclusions can be further strengthened.

All the other issues have been addressed in the revision.

I note that in the "major revision" reply in the system, there is a phrase "control missing in some experiments" - that it not the problem here, controls look adequate.

Author Response

Dear Reviewer,

Thank you very much again for your constructive comments and that your efforts to discuss critical issues related to our manuscript.

Concerning your first point of critique about the combination of datasets:

We still believe that figures 5, 6 and 7 with different levels of data merging are needed to make our point on genetic variation as a factor that adds significant noise to this type of experiments (e.g. the often neglected differences between individual genotypes, see Fig 7). Although some of the differences you mentioned are not significant all treatments show significant differences for some of the genotypes in both the resistant and the susceptible group.

In contrast, figures 5 and 6 (the effect of individual and combined stresses across all genotypes in figure 5 and specific effects of the Rdr1 based resistance or it absence in figure 6 dividing all genotypes into resistant and susceptible genotypes) are needed to show that there are general, overall effects (figure5) and that extent of effects differ between resistant and susceptible genotypes.

Concerning your critique on the reference point for normalisation:

We demonstrated effects that stress factors have on gene expression. For this we think that the unstressed control (full nutrition= no phosphate stress and inoculation with water = not pathogen stress) is the proper control for all of these effects. This is a standard procedure for experiments on the effects of stress factors. Using a control with one stress factor would make it impossible to analyse combined effects or the effect of this particular stressor.

Concerning your point on potential technical problems in figure 7 row 3 for the normalisation

Normalisation of the unstressed water controls in figure 7 was done by using one additional biological repeat as the reference for each genotype.

However, we agree that normalisation in the old version of figure 7 was not well explained and think that the data on unstressed plants did not contribute much to the point we wanted to make on individual variation. So we changed figure 7 in deleting the column with unstressed water controls under full nutrition. The other treatments are normalised to the values of the unstressed controls.

Here, we also added some information to materials and methods to explain our procedure of normalisation more clearly to the reader.